# Ultra-Wideband Radar for Simultaneous and Unobtrusive Monitoring of Respiratory and Heart Rates in Early Childhood: A Deep Transfer Learning Approach

**DOI:** 10.3390/s23187665

**Published:** 2023-09-05

**Authors:** Emad Arasteh, Esther S. Veldhoen, Xi Long, Maartje van Poppel, Marjolein van der Linden, Thomas Alderliesten, Joppe Nijman, Robbin de Goederen, Jeroen Dudink

**Affiliations:** 1Department of Neonatology, University Medical Center Utrecht, Wilhelmina Children’s Hospital, 3508 EA Utrecht, The Netherlands; e.arastehemamzadehhashemi@umcutrecht.nl (E.A.); m.roos1@students.uu.nl (M.v.d.L.); t.alderliesten-2@umcutrecht.nl (T.A.); robbindegoederen@gmail.com (R.d.G.); 2Department of Electrical Engineering (ESAT), STADIUS Center for Dynamical Systems, Signal Processing and Data Analytics, KU Leuven, 3001 Leuven, Belgium; 3Pediatric Intensive Care Unit and Center of Home Mechanical Ventilation, University Medical Center Utrecht, Wilhelmina Children’s Hospital, 3508 EA Utrecht, The Netherlands; e.s.veldhoen@umcutrecht.nl (E.S.V.); m.vanpoppel@umcutrecht.nl (M.v.P.); j.nijman@umcutrecht.nl (J.N.); 4Department of Electrical Engineering, Eindhoven University of Technology, 5612 AE Eindhoven, The Netherlands; x.long@tue.nl

**Keywords:** early childhood, heart rate, monitor, respiratory rate, UWB radar

## Abstract

Unobtrusive monitoring of children’s heart rate (HR) and respiratory rate (RR) can be valuable for promoting the early detection of potential health issues, improving communication with healthcare providers and reducing unnecessary hospital visits. A promising solution for wireless vital sign monitoring is radar technology. This paper presents a novel approach for the simultaneous estimation of children’s RR and HR utilizing ultra-wideband (UWB) radar using a deep transfer learning algorithm in a cohort of 55 children. The HR and RR are calculated by processing radar signals via spectrogram from time epochs of 10 s (25 sample length of hamming window with 90% overlap) and then transforming the resultant representation into 2-dimensional images. These images were fed into a pre-trained Visual Geometry Group-16 (VGG-16) model (trained on ImageNet dataset), with weights of five added layers fine-tuned using the proposed data. The prediction on the test data achieved a mean absolute error (MAE) of 7.3 beats per minute (BPM < 6.5% of average HR) and 2.63 breaths per minute (BPM < 7% of average RR). We also achieved a significant Pearson’s correlation of 77% and 81% between true and extracted for HR and RR, respectively. HR and RR samples are extracted every 10 s.

## 1. Introduction

Unobtrusive home screening methods for heart rate (HR) and respiratory rate (RR) during sleep in a pediatric population may be preferred over hospital-based monitoring for several reasons [1]. Firstly, home monitoring is often more convenient and less disruptive to daily routines than hospital-based monitoring. Secondly, it can be a less expensive screening tool than hospital-based monitoring, making it more cost-effective [2]. Thirdly, children and parents may be more relaxed and comfortable in their own homes, leading to more reliable monitoring results (in terms of reflecting real free-living conditions) [1]. Fourthly, families may be more likely to comply with home monitoring protocols than with hospital-based monitoring, which can result in more consistent and continuous data (compared to the spot-check in the hospitals) [3]. Lastly, home monitoring may reduce the risk of (hospital-acquired or patient-transferred) infections, which can be a concern [1,2]. Consequently, the need for safe and reliable non-invasive and non-contact physiological measurement continues to grow, and technologies providing this type of data are gaining more attention from healthcare researchers [4,5]. Among the important physiological signals to capture without direct contact, HR and RR measurement has garnered considerable interest in monitoring health in pediatric patients [6,7]. Various unobtrusive home monitoring systems, including actigraphy, video, and ballistocardiograph, have been used to overcome the limitations of monitoring via skin sensors [8]. One appealing technique for contactless RR and HR estimation is the utilization of ultra-wideband (UWB) radar technology for capturing vital signs. UWB radar is a method that does not require direct contact, and compared with video technology, it does not require ambient light, as videos without sufficient illumination for RR and HR estimation can lead to inaccurate results [9]. UWB radar has been demonstrated to be capable of accurately monitoring RR and HR, even in the presence of movement, with a precision similar to traditional contact-based methods [10,11,12]. This pulse-based radar system is capable of non-invasive vital sign measurement and can distinguish between different objects with high specificity. UWB radar can detect human presence and movement up to a distance of 10 m [13]. Despite its potential, the use of UWB radar for measuring HR in a pediatric population, particularly for children whose HRs are higher and more variable than adults, might be limited, mostly due to the lack of validation studies.

Previous studies have established a correlation between the peak frequency component of radar signals and heart and HRs [14]. However, in most research dealing with the young age range of patients, the cardiorespiratory features of patients are primarily utilized for sleep stage classification rather than directly determining HR and RR through radar [15]. This can be attributed to the intra-individual and inter-individual variability in HR and RR of young children. In addition, infants can exhibit a sleep-related respiratory pattern called “periodic breathing,” in which the amplitude of their tidal breathing is modulated across respiration. This periodic modulation results in the generation of harmonics, making conventional monochromatic frequency extraction an inadequate method for accurate RR and HR estimation [16]. This can lead to inaccurate diagnosis and management of cardiorespiratory disorders in infants. To address this issue, more sophisticated methods (than just finding the dominant frequency of radar signal) are necessary for accurate RR and HR estimation using radar in infants with periodic respiratory patterns.

There are studies that have already incorporated the latest machine learning and neural network approaches for extracting vital signs (which will be discussed in detail in the Section 2). However, one common drawback of almost all of these methods is the small biomedical data size compared to what is needed for training big phase space variables of learning networks [17].

One approach to overcome the use of a relatively small data set (in hospitals, intensive care units (ICUs), and home monitoring) is to apply transfer learning [18]. In the area of machine learning, transfer learning has been traditionally employed to overcome the issue of domain gap when utilizing previously gained knowledge on training data (source) for solving distinct problems on different test (target) data. This approach has been recognized as a solution for dealing with the challenge of insufficient training data [19]. It has been demonstrated that the primary layers of a Deep Convolutional Neural Network (DCNN) trained on a vast data set can extract generic features [20,21]. One favorable network in literature for transfer learning is the VGG-16 architecture [22] trained on the ImageNet dataset pre-trained model [23]. It has already been shown that the VGG-16 network is able to extract informative features for vital signs (HR/RR) estimation among adults [24].

Our study aims to accurately estimate HR and RR from UWB radar signals. This contactless method can provide valuable information about RR and HR disturbances in children outside a clinical setting. To check whether RR and HR in children can be efficiently carried out by a remote and non-invasive method, we conducted a mono-center prospective observational study in children undergoing polysomnography (PSG) to assess the potential of home monitoring as a screening tool for respiratory and HR disturbances. Our research presents a novel method for unobtrusively monitoring children’s heart rate (HR) and respiratory rate (RR) during sleep using ultra-wideband (UWB) radar. Unlike common traditional single-vital-sign approaches, we employ deep transfer learning with a pre-trained Visual Geometry Group-16 (VGG-16) model to concurrently estimate HR and RR. Incorporating spectrogram analysis, we convert radar signals into 2-dimensional images, enhancing our approach’s uniqueness and potential for accurate pediatric vital sign monitoring.

## 2. Latest Literature Survey

Some studies have recently explored the use of neural networks to extract vital signs from radar data in a contactless manner. Wu et al. [25] used a convolutional neural network to track and analyze the individualized skin displacement of targets to extract HR. However, this approach is not practical for subjects who are constantly moving during different states of sleep, eating, and other types of activities in their home environment.

In [26], a long short-term memory network (LSTM) approach is proposed to extract HR for adults, which works well based on a motion and distortion correction method named as Eclipse Fit method. However, this type of motion correction performance degrades in high levels of noise (which could possibly be the case for hospital and home environments and the various uncontrolled clutters present there). Furthermore, due to the common problem of vanishing gradient [17], a previously trained LSTM on a limited number of subjects cannot be considered a reliable and generalizable solution for hospitalized environments (especially with limited data available after data exclusion [27]).

In [28], a non-contact pediatric respiratory rate monitor (PRR-Monitor) was developed using a 24 GHz microwave radar to accurately determine respiratory waveform and heart rate (HR) in a 15 s interval. The algorithm, “Alternate Distinguishing Inhalation from Exhalation” (ADIE), effectively distinguishes inhalation and exhalation cycles using radar dual outputs (in-phase and quadratic), leading to precise thoracic motion velocity extraction and subsequent respiratory waveform derivation. While showing promise for non-contact RR and HR measurements in pediatric care, the PRR-Monitor’s accuracy may be influenced by patient motion artifacts during measurements, warranting careful consideration.

In [29], a robot-mounted millimeter-wave (mmWave) radar is employed to track HR changes based on the daily activity poses and movements of subjects through updates to neural network weights. Nevertheless, this method requires a big training dataset, which is not feasible either in hospitals (due to large inevitable data exclusion) or in home (individual) monitoring.

In recent developments, the potential of millimeter-wave frequency-modulated continuous wave (FMCW) radar for remote heart rate measurement has gained attention. A recent study by Jung et al. [30] introduces a novel approach that leverages the frame structure of FMCW radar systems to reduce measurement time for remote heart rate measurement. By adopting multiple sampling rates within fixed frame intervals, this method demonstrates improved resolution in heart rate measurement within a short timeframe. This method assumes minimal random body movement for stable phase changes between chirp signals, potentially limiting its accuracy in real-world scenarios with subjects exhibiting movement. Additionally, the trade-off between accuracy and computational complexity should be considered when increasing chirps and idle time to enhance frequency bin resolution.

Shi et al. [31] address the challenge of extracting heartbeat information from weak thoracic mechanical motion in noncontact vital sign measurement using Doppler radar-based applications. The proposed method integrates STFT, SVD, and ANC techniques and is validated using simulated and laboratory data for heart rate detection and variability in rest states. Despite its effectiveness, the sliding window application introduces time delay limitations compared to some alternatives. This approach has not been tested on early childhood data in a hospital setting, offering an advantage in applications with flexible delay requirements. While the Doppler radar biosensor provides convenient noninvasive measurement, its performance may not surpass existing methods in scenarios with high delay efficiency demands. Table 1 lists the main related studies to our research.

## 3. Materials and Method

### 3.1. Study Population

All recordings and evaluations were conducted exclusively at the Wilhelmina Children’s Hospital in Utrecht, the Netherlands. The data were collected over the period spanning from February 2021 to October 2022. In order to ensure accuracy and maintain a certain level of homogeneity within the study population, certain exclusion criteria were established, including, but not limited to, overt brain injury (such as intraventricular hemorrhage greater than grade 2) and congenital abnormalities (at the time of recording). Prior to the enrollment of participants, written informed consent was obtained from the parents. Moreover, permission to use pseudonymized patient data was granted by both the parents through the consent form and the local Medical Ethical Review Committee (METC number 21-816/C). Table 2 provides both general patient characteristics and specific characteristics observed during radar and vital sign recordings.

Recordings were made utilizing either an Intel^®^ RealSense Depth Camera D435i camera (Intel Corporation, Santa Clara, CA, USA) or a 1SEE VDO360 camera (VDO360, Maryland, USA) both of which captured at 30 frames per second, with a 1920 × 1080 pixel resolution. Additionally, a Xethru X2M200^®^ radar module (Novelda AS, Oslo, Norway) was employed (technical specifications are discussed in Section 3.2). The video camera and the UWB radar were both attached to an independent laptop next to the bed. The main factors of interference were interference with parents who slept in the bed next to the bed of the child and interference with the nurse who checked patients during the night, which were recognized by the camera and counted as excluded data (radar) segments.

Figure 1 depicts the general schematic approach applied in this paper. It starts with simultaneous video, radar, and vital signs recordings. After that, proper joint analysis of radar and video data helps us to omit the epochs of radar data contaminated with excessive movement. The data without dominant movement and artifacts are then transformed into 2D images for feeding into the neural network. True HR/RR tags are already saved and fed into the network as a response at the same time. Finally, the model performance (both inter- and intra-subject approaches) is evaluated.

### 3.2. Radar Data Acquisition

The X2M200 radar module (used for recording in this study) is capable of detecting ranges within 0.5 to 2.5 m across 52 bins, with a sensitivity range between 0 and 9 [32,33]. A higher sensitivity enables the identification of smaller objects but increases the likelihood of false alarms. The radar transmits pulses in two frequency bands, specifically 6.0–8.5 GHz and 7.25–10.2 GHz. Every bin generates baseband data 20 times per second. Additionally, the radar antenna features a front-to-back ratio of over 14 dB and a 7dB gain. This study utilized 52 bins, each with a length of 0.0388 m. In addition, a carrier frequency of 7.46 GHz, a detection zone spanning 0.4 to 1 m, and a range offset of 0.3 m were employed. The sensitivity level for the subjects’ recordings was either 5 or 9.

### 3.3. Data Inclusion

The data recorded in the Pediatric Intensive Care Unit (PICU) for patients under constant treatment are subject to issues, such as clutter and noise. In our view, this can be a proper upper bound of artifact for noisy and crowded environments of home settings (mainly due to blocking of signal transmission (occlusion of radar)). In this manner, by proper pre-processing of PICU data, we can take the first step in dependable pre-processing of home screening for RR and HR estimations. To determine the optimal threshold for data exclusion, we compared the movement computed from the radar data to the corresponding patient video. We reviewed 293 h of video-recorded data for the subjects. Our heuristic analysis showed that, for each value of 10 s movement of motion signals (generated by summation of the differences between two consecutive time frames across the amplitude baseband signals), movement values (the mentioned summation over 10 s) less than 300 accurately indicate the absence of a significant clutter. Through a 5 min time window, if there were more than 10 epochs (of 10 s length) with movement of more than 300, the 5 min time window has been discarded.

To validate this observation, we analyzed sudden drops and falls in the grayscale intensity (average pixel) changes of the video frames; as an example, refer to Figure 2. It is obvious that there is a clear pattern of increase in radar movement values and changes of intensity pixels of video (changes in the scene). Further details on the relationship between the video pixel intensity changes and the radar movement values for the whole synchronized 293 h are given in the Appendix A, i.e., in Appendix A.

### 3.4. Pre-Processing and 2D Image Generation from Radar Data

The initial dataset consisted of 98 subjects with synchronized radar and PSG data (including HR and RR measured with electrocardiography, and thoracic impedance and thoracic belt). The ECG and thoracic impedance are obtained from the patient monitor (Philips Interlevel MP70 or MX800, from the Netherlands).

After data exclusion, there were a total of 55 subjects whose data were acceptable to be used based on the mentioned movement threshold of 300. Finally, we had a total of 68 h of usable segmented epochs from these 55 subjects. We applied RR and HR estimation based on these epochs, considering each as an HR sample to be estimated. As true RR and HR data are recorded every 0.33 s (3 Hz from the patient monitor), for true HR, we used the average of each 30 samples (for each epoch).

In this study, the frequency spectrum magnitudes (of each radar bin) for a 10 s duration, covering a range of frequency components from 0.2 to 4 Hz, are utilized (144 frequency samples). By selecting bins with the least range difference from the target, images of 144 × 177 dimensions are formed (177 time samples for the 3 most variable bins (based on the distance of the subject)). A pre-trained VGG-16 network, previously trained on the ImageNet dataset, is leveraged to enhance sensitivity to changes in image patterns (resized to 224 × 224). This transfer learning approach enables the implementation of a deep neural network model (DNN) on medical data, despite the limited sample size compared to other DNN fields of research, such as natural language processing and computer vision. One example of the spectrum image (input to model) for 10 s epoch of radar data is shown in Figure 3A.

### 3.5. Transfer Learning and Model Definition

The aim of transfer learning is to address the issue of domain gap, where the knowledge obtained from bigger data (source) is used on different smaller (target) data to tackle the problem of insufficient data for training in the smaller dataset. This approach is necessarily helpful in our study because of the small size of the dataset and subjects might likely have significant variability in demographic and possibly physiological characteristics due to the presence of a big age range.

The initial layers of a DCNN, trained on a large dataset, can capture general features [22]. By utilizing deep transfer learning on a large dataset, we can fine-tune a pre-trained model on our (small) dataset.

In our study, we utilized the VGG-16 architecture [23], which was pre-trained on the ImageNet dataset, including 1.2 million images. We only used the weights of convolutional layers from the pre-trained model. To tailor the model for our regression task in estimating HR and RR from radar signals, we made modifications to the VGG-16-based model, which is shown graphically in Figure 3 (and spectrograms for four different sets of HR/BR magnitudes are shown in Appendix A). The same modification was applied in two different branches for simultaneously extracting HR and RR. These modifications included the following:**(A)** adding 2D convolutional layer ((filter width × filter height) × number of neurons) of ((3 × 3)× 16) and a max-pooling layer (filter width × filter height)× of (3 × 3).**(B)** developing a fully connected layer (width × height × depth) of (1 × 1 × 8) with the activation function of “Relu” and another one with (1 × 1 × 4) with the activation function of “Relu”.**(C)** applying the fully connected layer (1 × 1 × 1) with the activation function of the “linear” type for the regression task for each branch of HR/RR estimation.**(D)** model compiled (for Keras model in Python) by Adam optimizer (learning rate = 0.005 and decay = 0.001).**(E)** loss function is mean square error and the batch size is 32.**(F)** for the fine-tuning, we only updated the last 5 convolutional layers’ weights (mentioned in A and B).

### 3.6. Model Evaluation Strategies and Statistics

We conducted a 10-fold cross-validation (10-fold CV) of the pooled data from the data of 55 subjects. We considered all samples of 68 h from all subjects in a pool, followed by a random split of the total samples into 10 folds. For each iteration of the CV, 9 folds of the data were further split into a training set (75%) for model training and a validation set (25%) for model optimization.

## 4. Results

The results of the study demonstrated the effectiveness of the proposed model in estimating RR and HR values. The evaluation was performed on a dataset consisting of 55 subjects out of the total 98 subjects, using 10-fold cross-validation.

The scatter plot in Figure 4A visualized the relationship between the estimated and real RR values for each of the 10 folds, while Figure 4B presented the Bland–Altman plot, providing insights into the accuracy of HR estimation. The same goes on for HR through Figure 5A,B.

The visual representation provided by Figure 4A offers an insightful perspective into the intricate relationship between the estimated and actual RR values for each of the 10 distinct folds. This scatter plot not only underscores the model’s performance across various instances but also offers a glimpse into the distribution and alignment of the estimations with the true values. Meanwhile, Figure 4B, illustrated as the Bland–Altman plot, presents a more detailed depiction of the accuracy in HR estimation. This visualization method, renowned for highlighting discrepancies and potential biases between measurements, affords a nuanced understanding of the model’s performance.

To present a succinct yet comprehensive quantitative overview of the 10-fold cross-validation outcomes, we have encapsulated the key results in Table 3. This tabular representation encapsulates the essence of the model’s performance in estimating both RR and HR values across various folds. Notably, the mean absolute error (MAE) ranges for RR estimation were between 2.28 and 2.79 breaths per minute (BPM), whereas for HR, the MAE varied slightly from 7.81 to 8.23 beats per minute (BPM). Pearson’s correlation coefficients, which reveal the strength and direction of linear relationships, consistently demonstrated robust dependencies. The correlation values for RR estimation ranged from 0.78 to 0.83, while for HR, they spanned from 0.76 to 0.79. Moreover, the calculated mean biases for both RR and HR estimation were relatively small in magnitude, fluctuating between −0.26 to 0.18 BPM and 0.21 to 0.55 BPM, respectively.

However, it is the limits of agreement (LOA) values that provide a more insightful understanding of the model’s performance. These values, denoted as the range within which deviations between estimated and true values lie, encompass 6.03 to 7.65 BPM for RR and 21.8 to 23.4 BPM for HR. This depiction of the acceptable range of discrepancies underscores the model’s reliability in practical applications.

Before delving into the analytical examination of our findings, we first outline the procedural groundwork that underpins these results. Our study centers on the estimation of heart rate (HR) and respiratory rate (RR) through the utilization of non-invasive ultra-wideband (UWB) radar signals.

The outcomes we present stand as a reflection of this rigorous process. We offer a visual representation, captured in Figure 4A,B, which unravels the intricate association between estimated and actual RR values across distinct folds. This visualization not only encapsulates our model’s performance but also offers valuable insights into its nuances.

Our quantitative summary, encapsulated in Table 3, distills the essence of our 10-fold cross-validation process. Noteworthy is the consistent mean absolute error (MAE) range for RR estimation, spanning 2.28 to 2.79 breaths per minute (BPM), alongside an analogous range of 7.81 to 8.23 BPM for HR estimation. These results emphasize the model’s ability to provide reliable estimates for respiratory and heart rates.

Pearson’s correlation coefficients emerge as key indicators, demonstrating robust linear relationships. Ranging between 0.78 and 0.83 for RR and 0.76 and 0.79 for HR, these coefficients underscore the model’s proficiency in capturing patterns amidst real-world noise.

Further affirmation is found in the examination of mean biases. With relatively modest biases oscillating between −0.26 and 0.18 BPM for RR and 0.21 and 0.55 BPM for HR, the model demonstrates its precision by offering accurate estimations with minimal systematic discrepancies.

Lastly, the practical reliability of our model comes into focus through the limits of agreement (LOA) values. These values—spanning 6.03 to 7.65 BPM for RR and 21.8 to 23.4 BPM for HR—elucidate the range of permissible deviations between estimated and actual values, accentuating the model’s applicability in real-world scenarios.

### 4.1. Analysis of the Results

The presented outcomes provide an illuminating perspective on the practical applicability of our proposed model for estimating heart rate (HR) and respiratory rate (RR) through the utilization of ultra-wideband (UWB) radar signals. These findings, combined with key concepts such as transfer learning and data pre-processing, highlight the model’s robustness in a healthcare context.

#### 4.1.1. Visual Representation: Insights through Visuals

Visual representation plays a pivotal role in enhancing our understanding of the intricate interplay between estimated and actual RR values across 10 distinct folds. Figure 4A’s scatter plot effectively captures the model’s performance variations across instances, while Figure 4B’s Bland–Altman plot goes further by providing deeper insights into the accuracy of HR estimation. This visualization approach unveils both the model’s strengths and limitations, making it an indispensable part of the analysis.

#### 4.1.2. Quantitative Summary: Key Metrics at a Glance

Moving beyond visuals, Table 3 offers a concise yet comprehensive encapsulation of the crucial outcomes obtained from the 10-fold cross-validation process. Mean absolute error (MAE) ranges are integral in evaluating the model’s consistency. For RR estimation, the MAE ranges from 2.28 to 2.79 breaths per minute (BPM), revealing the model’s ability to provide reliable estimates of respiratory rates. Correspondingly, in the case of HR estimation, a slight MAE variation of 7.81 to 8.23 BPM underlines the model’s proficiency in accurately estimating heart rates.

#### 4.1.3. Robust Dependencies: Pearson’s Correlation Coefficients

A hallmark of the model’s effectiveness is the consistent and robust linear dependencies shown by Pearson’s correlation coefficients. These values, ranging from 0.78 to 0.83 for RR estimation and 0.76 to 0.79 for HR estimation, emphasize the model’s capacity to capture patterns within vital sign data. Despite the inherent noise in real-world measurements, these high correlation values validate the model’s ability to discern underlying trends.

#### 4.1.4. Addressing Systematic Errors: Mean Biases Examination

Examining mean biases adds another layer of validation to the model’s performance. The relatively minor magnitude of biases, ranging from −0.26 to 0.18 BPM for RR and 0.21 to 0.55 BPM for HR, indicates the model’s competence in delivering accurate estimates without introducing substantial systematic errors.

#### 4.1.5. Limits of Agreement: Gauging Practical Reliability

However, it is the limits of agreement (LOA) values that provide a comprehensive understanding of the model’s practical reliability. These values, encompassing 6.03 to 7.65 BPM for RR and 21.8 to 23.4 BPM for HR, illustrate the acceptable range of deviations between estimated and true values. This dimension underscores the model’s applicability in real-world scenarios, where small discrepancies are deemed acceptable.

#### 4.1.6. Methodology: Integrating Data for Model Performance

Crucial to our methodology is the joint analysis of radar and video data to ensure data quality. This step is essential to creating a dataset well-suited for neural network processing. The introduction of a pre-trained VGG-16 model, originally tailored for image recognition, effectively captures intricate radar data patterns. This exemplifies the adaptability of deep learning concepts to medical contexts.

In summation, the comprehensive results eloquently align with our study’s objectives. The proposed model showcases robust linear dependencies, minimal errors, and reasonable limits of agreement, underscoring its applicability across diverse healthcare contexts. These findings significantly contribute to the advancement of medical monitoring practices, ultimately enhancing the standards of patient care. With these outcomes, our study lays a solid groundwork for future developments in medical technology, highlighting the potential of UWB radar and deep learning for precise pediatric vital sign estimation.

## 5. Discussion

The main findings of this study demonstrate that the use of UWB radar and a deep transfer learning algorithm can accurately and unobtrusively monitor children’s HR and RR during sleep. Our study introduces an innovative approach for the simultaneous estimation of children’s heart rate (HR) and respiratory rate (RR) during sleep via ultra-wideband (UWB) radar. Utilizing deep transfer learning with a pre-trained Visual Geometry Group-16 (VGG-16) model, our method stands out in its ability to capture both HR and RR trends accurately. Spectrogram analysis further distinguishes our research by translating radar signals into 2-dimensional images. Covering an age range of 13 days to 18 years, our approach accommodates diverse physiological characteristics, enhancing its versatility. Despite challenges like motion artifacts and subject variability, our method maintains robust performance. Comparative evaluations against existing studies underscore its superiority in accuracy, correlation, and bias metrics for HR and RR estimation, even when dealing with abnormal polysomnography (PSG) results. Beyond pediatric care, our approach offers broader applications in remote patient monitoring and sports performance tracking, expanding its potential impact. Overall, our work provides an innovative framework for accurate and unobtrusive pediatric vital sign monitoring, with implications for early issue detection and improved healthcare practices. In general, the model follows the RR signal trend better with a little higher error. The proposed method (according to Figure 4) achieved a mean absolute error of 8.04 BPM and a significant correlation of 81% for HR and a mean absolute error of 2.63 BPM and a significant correlation of 77% for RR, with an average error in prediction of at most 7% of the true vital sign values. This suggests that the proposed approach could be a valuable tool for early detection of potential health issues in children, as well as reducing unnecessary hospital visits.

Comparing our results to prior literature, we note that there are very few studies that remotely extract HR every 10 s for the age range of our dataset, and most of them exclude motion artifact data. One proper example is the study of Al-Naji et al. [21], which uses a hovering unmanned vehicle to extract RR. However, big devices like this (if used for radar applications) take advantage of a significantly big synthetic radar aperture which is not feasible either in PICUs or home applications. Masagram et al. [34] extracted the vital signs of 12 human adults using pulse Doppler radar, with a mean accuracy of up to 75% of average HR for subjects with artifacts, while our worst result is less than 6% of average HR. Yoo et al. [35] used convolutional neural networks and GoogLeNet to extract vital signs for an age range of up to 13 years, with a reported mean bias (of Bland–Altman) of 1.8, while our reported mean bias is less than 0.4. Kim et al. [36] achieved an upper limit (of Bland–Altman) for RR estimation equal to 14, which is higher than our reported upper limit of <7.4. Moreover, they have applied denoising based on only six neonate subjects, which makes their pre-processed data much more dependable on subject-specific characteristics. Katoh et al. [28] extracted RR by comparing the in-phase and quadratic components of the down-converted signal, which resulted in a mean value of 0.61 BPM, in which our results were superior in this aspect. However, this research achieved a very high correlation between true and extracted RR while the patients were sitting in a fixed place. As a result, this correlation would be different if it had been computed for cases with freedom of movement, like our study. The MAE of heart rate in [30] reaches a promising value of less than 5 BPM; however, this is only applied to 3 subjects with natural health status, and the results will definitely be different for a hospitalized environment. Although the method of Shi et al. [31] is innovative, there is no report of detailed examination of relations between true and extracted heart rate (like correlation, MAE, and LOA). Furthermore, it is used for a fixed sitting subject, which is not feasible for either ICU or home monitoring applications.

Our research presents a notable advancement over previous related studies due to its distinctive ability to simultaneously extract heart rate (HR) and respiratory rate (RR) with exceptional accuracy across an extensive age range encompassing individuals from just 13 days old up to 18 years. Notably, this age range represents a period of significant physiological variability and encompasses subjects who predominantly exhibit abnormal polysomnography (PSG) results. This aspect differentiates our work from the existing literature, where similar endeavors often focus on narrower age ranges and, in some instances, exclude subjects with abnormal PSG results.

To exemplify this advantage, we refer to the study by Yoo et al. [31], which combined subjects across all age ranges from 0 to 13 years. Interestingly, their findings indicate that such a wide age span led to a marked reduction in prediction accuracy, dropping from 92% to 58.5%. In contrast, our approach remains resilient to the challenges posed by subject variability, as evidenced by consistently high accuracy across a diverse age spectrum. This robustness not only underscores the versatility of our model but also positions it as a promising solution for capturing the nuances of physiological variations in pediatric populations.

Central to our research is the development of an innovative method for unobtrusively monitoring the HR and RR of children during their sleep, facilitated by ultra-wideband (UWB) radar technology. Based on our knowledge, this is the first pediatric research for vital signs extraction of hospital data through deep transfer learning. What sets our approach apart is its departure from conventional single-vital-sign paradigms. Instead, we employ deep transfer learning, harnessing the prowess of a pre-trained Visual Geometry Group-16 (VGG-16) model to estimate both HR and RR concurrently. Notably, this methodology is enriched through the integration of spectrogram analysis, which facilitates the transformation of radar signals into 2-dimensional images. This distinctive feature enhances the accuracy and potential of our pediatric vital sign monitoring approach. Considering the limitations inherent in existing methodologies, our approach rises to the forefront. The novelty of our method circumvents challenges observed in other techniques. Traditional single-vital-sign strategies may falter in the face of the intricate dynamics of pediatric vital signs. The introduction of deep transfer learning addresses this limitation, as it leverages a model that has been trained on a vast array of images. Unlike methods burdened by constraints related to motion artifacts or reliance on large training datasets, our method synthesizes advanced technology, transfer learning, and spectrogram analysis to provide a comprehensive solution to the challenges posed by pediatric vital sign monitoring. Thus, the advantage of our method is firmly rooted in its capacity to transcend the limitations inherent in existing methodologies, thereby advancing the field of pediatric vital sign assessment.

In light of our findings, the implications for unobtrusive monitoring of children’s vital signs during sleep are significant. The proposed method emerges as an effective and reliable screening tool for potential health issues. Its non-invasive, continuous, and convenient nature holds the promise of enabling early detection of health problems and enhancing communication between caregivers and healthcare providers. This advancement could potentially alleviate the burden of unnecessary hospital visits and contribute to improved patient outcomes.

Lastly, while our study’s primary focus lies in unobtrusive monitoring of children’s vital signs during sleep, the utility of the proposed method extends beyond this scope. Its potential applications in remote patient monitoring and sports performance tracking are noteworthy. In these contexts, where accurate and continuous monitoring of vital signs is paramount, our approach’s precision and convenience could usher in new paradigms for healthcare and performance optimization.

In conclusion, our research not only demonstrates the capabilities of our proposed model but also showcases its advantages over previous approaches. By encompassing a wide age range and addressing subject variability, our method holds the potential to significantly impact pediatric healthcare and beyond. The avenue for future research is promising, with the potential for broader applications and real-world impact.

### 5.1. Limitations

Despite the promising results, there are several limitations that must be taken into account while using UWB radar to extract vital signs in the home environment. One of the significant limitations is the accuracy of measurements (observable in Appendix A). The radar signals are susceptible to noise due to factors, such as environmental conditions, motion artifacts, and interference from other devices. This noise can affect the accuracy of the measurements. Another limitation is the range of measurement. The accuracy of the radar signal decreases as the distance between the subject and the radar increases. Therefore, this technology may not be suitable for measuring physiological signals in subjects who are far away from the radar. Moreover, the physiological state of the subject can also affect the accuracy of the measurements. For example, movements and postural changes can lead to fluctuations in HR and RR, which can be challenging to differentiate from noise in the radar signal. Additionally, the use of radar technology to extract physiological signals may not be appropriate for all populations. Individuals with specific medical conditions or implanted devices may be at risk for adverse effects from the use of radar technology.

### 5.2. Future Perspectives

The use of the VGG16 network to extract vital signs from radar signals for a home screening application is a promising approach with significant potential for future research. This potentially reduces costs and is more patient-friendly.

In navigating the complex relationship between respiratory rate (RR) and heart rate (HR) measurements, our study employing meticulous temporal data segmentation, deep transfer learning with a pre-trained VGG-16 model, and a spectrogram-based strategy has demonstrated efficacy even amidst correlation, as indicated by mean absolute error and Pearson’s correlation. In future research, we aim to delve deeper into the temporal aspects of this correlation using techniques like cross-correlation analysis, addressing nuanced challenges and refining our approach, potentially integrating multimodal data sources for a more comprehensive insight.

Future work could explore the use of other deep learning models to enhance the accuracy of the measurements and compare their performance with that of VGG16. The selection of VGG-16 as our deep learning architecture stems from its appealing attributes, including its interpretable architecture with 16 consistent weight layers, making it conducive to transparent model behavior explanation. Its demonstrated transfer learning capability from ImageNet, apt for fine-tuning specific tasks with limited data, proves advantageous for medical applications where data scarcity prevails. VGG-16′s balanced performance, resource efficiency, and proven track record in various domains further bolster its reliability. As we move forward, these advantages provide a strong foundation for our planned future research, which will explore more advanced transfer learning techniques and models, guiding the evolution of our work in unobtrusive child health monitoring and related endeavors. Additionally, the use of radar signals to extract vital signs in other clinical settings could be explored, such as in adult intensive care units, neonatal intensive care units, or emergency departments. The development of a robust and accurate radar-based vital sign monitoring system could have significant clinical applications, particularly in situations where traditional methods are either impractical or not available. This technology could be particularly beneficial for patients with limited mobility, such as those in intensive care units or those with chronic illnesses. Overall, this study opens up new avenues for research on the use of deep learning models and radar signals to extract vital signs in various clinical settings, and it is expected that further studies will be conducted to explore its full potential. More research is required to facilitate easily accessible and safe non-contact methods for respiration monitoring in different real home settings to have a better understanding of noise, clutter, and probability density functions for radar assessments.

## 6. Conclusions

In this paper, we have shown that through a proper data exclusion (with appropriate threshold for radar movement) we can simultaneously estimate the true RR and HR of early childhood patients using data from UWB radar with high accuracy. This can be the first step towards a home screening method of vital sign measurement in a contactless manner.

## Figures and Tables

**Figure 1 sensors-23-07665-f001:**
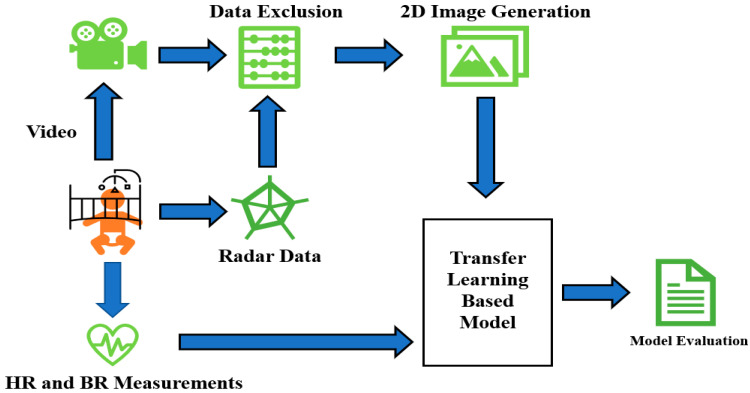
Schematic general overview of sleep stage classification steps in this research.

**Figure 2 sensors-23-07665-f002:**
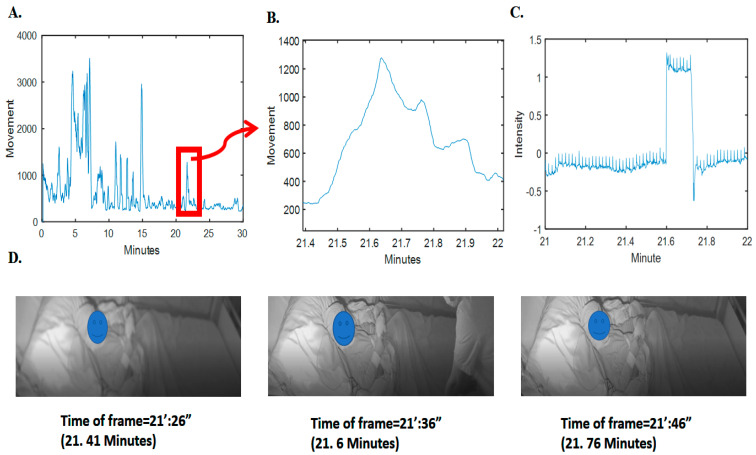
Movement computed from radar and the corresponding video signal intensity and frames. (**A**) Movement computed and depicted for 3 min of radar data. (**B**) The specified time samples for movement. (**C**) The corresponding grayscale intensity (normalized magnitude as pixel value subtracted by average value divided by standard deviation) in the specific time samples of video. (**D**) Three frames before, during, and after clutter (nontarget person) presence. For privacy issues, the face of the patient has been blocked.

**Figure 3 sensors-23-07665-f003:**
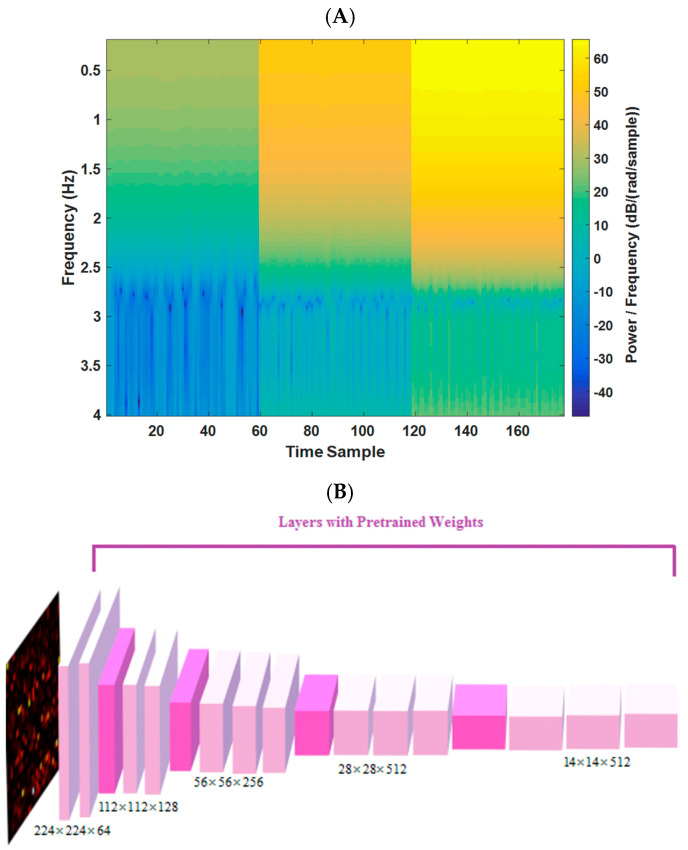
(**A**) Example of a 10 s spectrogram for one estimated tag. (**B**) The VGG-16 trained network architecture in this study (from [23]).

**Figure 4 sensors-23-07665-f004:**
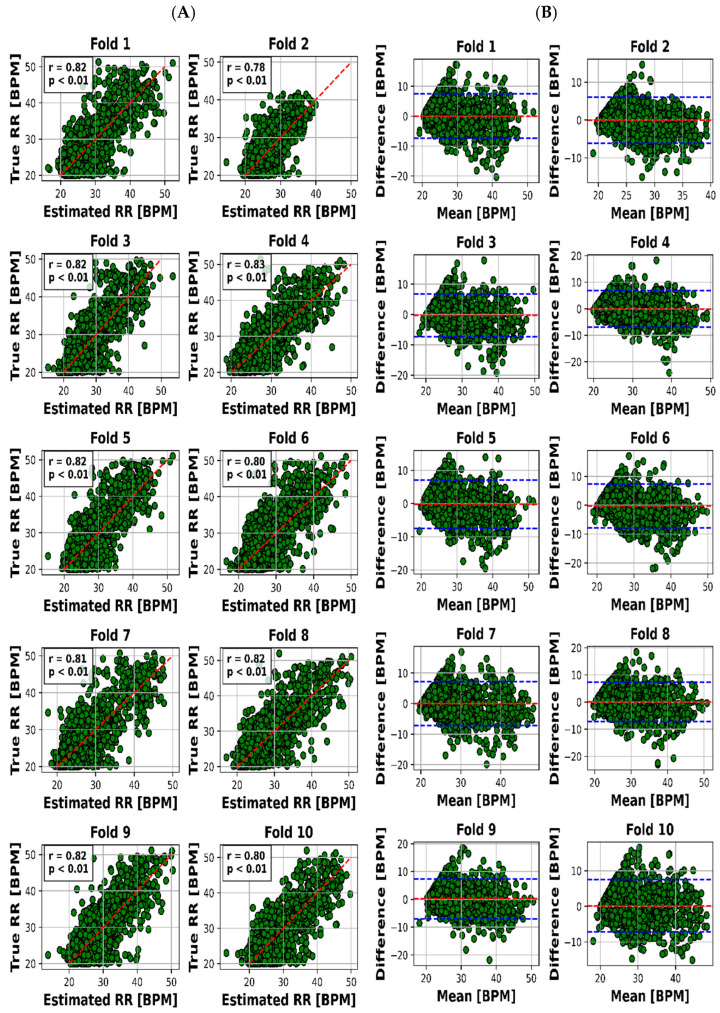
Results for RR estimation of 10 folds. (**A**) “True” versus “Estimated” and (**B**) Bland–Altman plot (blue lines are the limits of agreement (LOA)).

**Figure 5 sensors-23-07665-f005:**
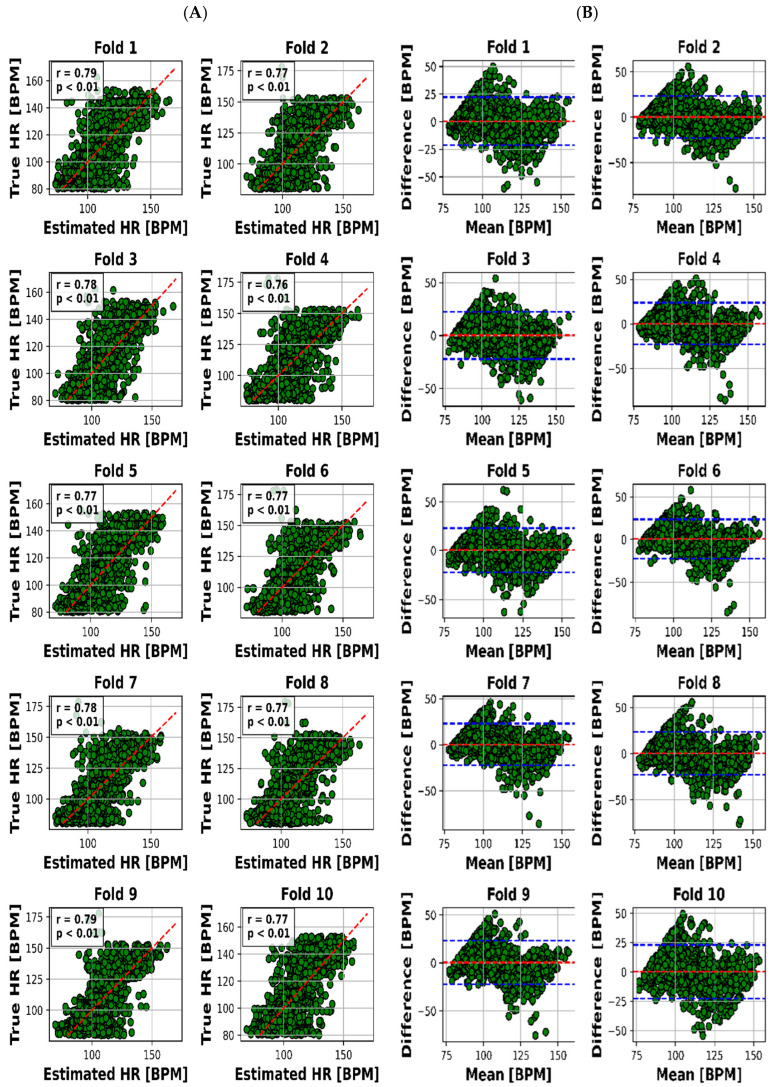
Results for HR estimation of 10 folds. (**A**) “True” versus “Estimated” and (**B**) Bland–Altman plot (blue lines are LOA).

**Table 1 sensors-23-07665-t001:** Summary of outputs, outcomes, and limitations of major references related to our study.

Reference	Outputs	Outcomes	Limitations
Wu et al. [25]	Individualized skin displacement for HR estimation	Estimation of HR using skin displacement	Movement-sensitive; limited to certain states of subjects
Han-Trong et al. [26]	HR estimation through LSTM network. Incorporates Eclipse Fit method for motion and distortion correction. Incorporates spectrogram analysis for enhanced accuracy	Improved HR estimation via deep transfer learning and incorporation of spectrogram analysis	Motion correction degrades in high noise; vanishing gradients limit generalizability and reliability
Katoh et al. [28]	Respiratory waveform and HR in 15 s intervals captured via radar. Effective use of “Alternate Distinguishing Inhalation from Exhalation” algorithm	Non-contact measurement of RR and HR in pediatriccare; accurate estimation of RR and HR	Susceptible to motion artifacts during measurements; requires careful consideration
Zhao et al. [29]	Heart rate changes tracked based on daily activity poses and movements. Incorporates mmWave radar on robot	Heart rate variations monitored through neural network weight updates; applicability in hospital and home environments	Demands substantial training dataset; infeasible for hospitals and home monitoring
Jung et al. [30]	Remote heart rate measurement using frame structure of FMCW radar systems	Improved heart rate measurement resolution within short timeframe; potential limitations in dynamic scenarios	Assumes minimal movement for stable phase changes; trade-off between accuracy and computational complexity
Shi et al. [31]	Heartbeat information extracted from weak thoracic mechanical motion via Doppler-radar-based applications	Detection and variability of heart rate through Doppler-radar-based methods; potential for non-contact vital sign measurement	Sliding window introduces time delay; not tested in pediatric data; may lack efficiency in scenarios with high delay; demands

**Table 2 sensors-23-07665-t002:** Patient characteristics.

N	55
**Mean age (range)**	6.1 year (10 day–18 year)
<1 year, N (%)	18 (33)
1–12 year, N (%)	29 (53)
>12 year, N (%)	8 (15)
**Male gender (%)**	32 (58)
**Mean birth weight in kg (range)**	3.1 (1.1–4.3)
**Apgar (appearance, pulse, grimace, activity, and respiration) score 1/5/10 min (N at 1 min/N at 5 min/N at 10 min)**	7.8/8.9/8.9 (21/22/9)
**Medication 24 h before or during observation, N (%)**	39 (71)
Caffeine	1
Doxapram	0
Hydrocortison (systemic)	1
**Syndromes/diagnosis**	
Spinal muscular atrophy type II	1
*Central nervous system*	*7*
Spina bifida aperta with Chiari malformation	2
Epilepsy	2
West syndrome	1
Joubert syndrome	1
Hydrocephalus	1
*Skeletal abnormalities*	*15*
Achondroplasia	12
Brachycephalia	1
Craniosynostoses	1
Arthrogryposis multiplex congenita	1
*Upper airway abnormalities*	*17*
Pierre Robin sequence	4
Palatoschisis	3
Laryngomalacia	2
Cheilognathopalatoschisis	2
Treacher Collins hypoplastic mandible	1
Midline facial cleft	1
Bifide uvula	1
Bronchomalacia	1
Tracheobronchomalacia	1
Laryngobronchomalacia	1
*Other syndromes*	*10*
Down syndrome	3
22q11.2 deletion syndrome	2
ROHHADNET (Rapid onset Obesity, Hypothalamic dysfunction, Hypoventilation, Autonomic Dysregulation and Neuroendocrine Tumor) syndrome	1
Carey Fineman Ziter syndrome	1
Kabuki syndrome	1
Leri Weill syndrome	1
Coffin Siris syndrome	1
*Premature/dysmature*	*9*
*No syndrome/unknown*	*13*

**Table 3 sensors-23-07665-t003:** Summary of quantitative results for RR/HR estimation through the 10-fold CV. MAE stands for mean absolute error between the true values and estimated ones for each fold. Corr (Correlation) is the Pearson’s correlation (linear dependency) between estimated and true values. Mean bias is the mean bias computed by the Bland–Altman figure. LOA (1.96 × standard deviation (true-estimated)) stands for a limit of agreement from Bland–Altman.

Fold No.	MAE for RR (BPM)	MAE for RR (BPM)	Corr for RR	Corr for HR	Mean Bias for RR (BPM)	Mean Bias for HR (BPM)	LOA for RR (BPM)	LOA for HR (BPM)
1	2.65	7.81	0.82	0.79	0.02	0.55	7.40	21.8
2	2.28	8.23	0.78	0.77	−0.04	0.30	6.03	23.4
3	2.55	7.94	0.82	0.78	−0.26	0.21	7.00	22.3
4	2.54	8.19	0.83	0.76	−0.06	0.52	6.93	23.4
5	2.72	8.04	0.81	0.77	−0.20	0.46	7.30	22.5
6	2.79	8.00	0.79	0.77	−0.24	0.54	7.65	22.8
7	2.61	8.05	0.80	0.78	−0.04	0.32	7.20	22.5
8	2.65	8.09	0.82	0.77	0.08	0.48	7.24	23.0
9	2.63	8.02	0.82	0.79	0.18	0.55	7.17	22.6
10	2.73	8.07	0.80	0.77	0.14	0.24	7.34	22.87

## Data Availability

The data cannot be shared due to confidentiality rules of children hospital.

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
