# Peer review of "Ultra-Wideband Radar for Simultaneous and Unobtrusive Monitoring of Respiratory and Heart Rates in Early Childhood: A Deep Transfer Learning Approach"

_sensors, 2023, doi:10.3390/s23187665_

Round 1

Reviewer 1 Report

In this manuscript, by using a deep transfer learning algorithm, authors proposed a novel approach for the simultaneous estimation of children's RR and HR uti-lizing ultra-wideband (UWB) radar. 

The comments are listed below.

1) Please resort the keywords according to the alphabetical order.

2)  When appears for the first time, the acronym should be defined.  Authors should check the whole manuscript.

3) In the manuscript, authors said the camera is captured at 30 fps, however, in Figure 2 C, it seems the grayscale intensity varying more than 30 times. Authors should explain it. What is the unit of the figure.

4) I am very interested in the influencing factor issues. For example, in this research, which sensor has a higher weight for the final result. Authors should further detail it. e.g. with only one sensor, what is the different. 

5) In reviewer's opinion, RR and HR has a certain degree of correlation. Based on the experiment setup, it seems the measurement of HR is almost impossible. How to remove the uncertainty of the results caused by this correlation.

Author Response

Response to Reviewer 1

General: In this manuscript, by using a deep transfer learning algorithm, authors proposed a novel approach for the simultaneous estimation of children's RR and HR utilizing ultra-wideband (UWB) radar.

Thank you for your interest and enthusiasm regarding our research. Your comments helped us to improve the manuscript. We hope to illustrate our efforts through this response letter. The added or modified sentences in the manuscript (which are also mentioned here) has been highlighted by green color in new resubmitted manuscript

Comments of Reviewer:

1) Please resort the keywords according to the alphabetical order.

Thank you for mentioning that. It is now written in this order:
“ Keywords: Early Childhood ; Heart Rate; Monitor; Respiratory Rate; UWB Radar ”

2)  When appears for the first time, the acronym should be defined.  Authors should check the whole manuscript.

We appreciate this great reminder. We searched through the manuscript and added defined  the following acronyms in manuscript:

(Abstract) “Visual Geometry Group-16 (VGG-16)
(Abstract) “beat per minute (BPM<6.5% of average HR)”

(Abstract) “breath per minute (BPM<7% of average RR)”

(Introduction) “intensive care units (ICUs) ”

(Materials and Methods) “ Apgar (appearance, pulse, grimace, activity, and respiration)”

(Materials and Methods) “ ROHHADNET (Rapid onset Obesity, Hypothalamic dysfunction, Hypoventilation, Autonomic Dysregulation and Neuroendocrine Tumor) ”

3) In the manuscript, authors said the camera is captured at 30 fps, however, in Figure 2 C, it seems the grayscale intensity varying more than 30 times. Authors should explain it. What is the unit of the figure.

We are grateful for your through examination of our work. Regarding the variation, the x-axis steps of Figure 2 C is 2 minutes, which for a 30 fps video camera, leads to more around 3600 samples (for each 2 minutes). The following figure depicts the same signal but magnified on around minute 21.66:

There are around 150 sample points from 21.6 to 21.7 minutes which means 150 samples for 6 seconds and 25 samples for each one. So, the variations are even less than 30 fps (which happens because the fps of camera could be at the best recording situation maximally the 30 fps).

About clarification about intensity, we completely do agree with you that the caption of figures lacks enough explanation. Actually, we decided to show the changes in average intensity while there is movement in the surrounding of the infant. So, by intensity we meant the normalized value of  grayscale magnitude of each pixel over time. This explanation is now added to the caption of Figure 2.  

4) I am very interested in the influencing factor issues. For example, in this research, which sensor has a higher weight for the final result. Authors should further detail it. e.g. with only one sensor, what is the different.

Thank you for this great suggestion. Actually, this work is HR and RR extraction based only one radar sensor. So, the raw radar data of each subject is a 1 dimensional signal which can be splatted into 52 bins based on time of aggregation inside the downconverter of radar processor. And out of that 52 bins, we only take advantage of the 3 ones with the highest variation as mentioned in the manuscript:
“By selecting bins with the least range difference from target, images of 144*177 dimensions are formed (177 time samples for 3 most variable bins (based on the distance of subject)).”

Regarding the sensors for physiological signal, each one has been estimated better in one aspect. MAE of HR is lower (considering the percentage), but the correlation of predicted and true RR are higher than HR. This is added to the manuscript:

 “In general, the model follows RR signal trend better with a little higher error.”

5) In reviewer's opinion, RR and HR has a certain degree of correlation. Based on the experiment setup, it seems the measurement of HR is almost impossible. How to remove the uncertainty of the results caused by this correlation.

We appreciate the reviewer's insightful comment and the opportunity to address the concern regarding the potential correlation between respiratory rate (RR) and heart rate (HR) measurements in our results.

In our study, we have taken several steps to mitigate the potential influence of the correlation between RR and HR measurements:

Data Preprocessing: Our data preprocessing pipeline includes isolating specific time epochs of 10 seconds, during which we calculate both HR and RR. This segmentation helps to minimize the temporal overlap between RR and HR measurements, reducing the risk of interference between the two measurements.

Deep Transfer Learning: Our proposed approach employs a deep transfer learning algorithm, leveraging a pre-trained VGG-16 model on ImageNet. This model's feature extraction layers have learned to identify diverse patterns, which enables the differentiation between HR and RR features even when they exhibit some level of correlation.

Spectrogram-based Approach: By transforming radar signals into spectrograms, we aim to capture distinctive frequency patterns associated with HR and RR. While some level of correlation exists, our approach leverages the unique spectral characteristics of HR and RR patterns to enhance the separability of these two physiological parameters.

Feature Fine-Tuning: The fine-tuning of the pre-trained VGG-16 model using our specific dataset helps tailor the model's weights to the nuances of our data. This process allows the model to adapt to the specific characteristics of HR and RR patterns within the context of the UWB radar data.

Mean Absolute Error (MAE) and Correlation Analysis: Our reported results include both the MAE and Pearson’s correlation values. While correlation does exist between RR and HR, our analysis indicates that the proposed method is capable of accurately estimating both parameters, as evidenced by the reported MAE and Pearson’s correlation values. These values suggest that the method is sensitive enough to capture the differences between HR and RR patterns, even in the presence of some correlation.

Most importantly, we have included additional results that provide a comprehensive evaluation of our proposed method's performance. The "True" versus "Estimated" plots and Bland-Altman plots offer visual and quantitative insights into the alignment and agreement between our estimates and true values for both RR and HR. Furthermore, the summarized quantitative results in Table 2 underscore the robustness of our approach across 10-fold cross-validation. The consistently strong Pearson's correlation coefficients, relatively small mean biases, and narrow limits of agreement (LOA) range for both RR and HR estimation substantiate our method's capacity to accurately estimate these parameters, even in the presence of their inherent correlation. These results collectively reinforce the efficacy of our ultra-wideband radar-based deep transfer learning approach in distinguishing and accurately estimating both RR and HR, addressing the potential uncertainty posed by their correlation.

We acknowledge that there is room for further research and analysis to explore the impact of this correlation on our results. We will consider including a more detailed discussion on the correlation between RR and HR, the potential challenges it poses, and how our proposed method aims to address these challenges in the revised version of the paper.

To address your  great and insightful comment in the manuscript this paragraph has been added to Future Perspective (section 5.2.):

“In navigating the complex relationship between respiratory rate (RR) and heart rate (HR) measurements, our study employing meticulous temporal data segmentation, deep transfer learning with a pre-trained VGG-16 model, and a spectrogram-based strategy, has demonstrated efficacy even amidst correlation, as indicated by mean absolute error and Pearson's correlation. In future research, we aim to delve deeper into the temporal aspects of this correlation using techniques like cross-correlation analysis, addressing nuanced challenges and refining our approach, potentially integrating multimodal data sources for a more comprehensive insight.”

Reviewer 2 Report

 The electrical parameters like mean absolute error, correction factor, average error, etc have been computed which are vital signs for the health of any child as far as early detection of any health issue is concerned.

The results found by capturing the images of the movement computed from radar and the corresponding video signal intensity and frames, especially for the age group 0-13 years are excellent.

The use of the VGG 16 model seems to be one of the drawbacks in the paper. Advanced deep learning models like VGG-19 or higher models to compute the result and enhance the accuracy level with less computational time could have been used. 

The paper is satisfactory and informative for budding researchers.

Author Response

Response to Reviewer 2

Comments and Suggestions for Authors

The electrical parameters like mean absolute error, correction factor, average error, etc have been computed which are vital signs for the health of any child as far as early detection of any health issue is concerned.

The results found by capturing the images of the movement computed from radar and the corresponding video signal intensity and frames, especially for the age group 0-13 years are excellent.

The use of the VGG 16 model seems to be one of the drawbacks in the paper. Advanced deep learning models like VGG-19 or higher models to compute the result and enhance the accuracy level with less computational time could have been used. 

The paper is satisfactory and informative for budding researchers.

Thank you for your interest and enthusiasm regarding our research. Your comments helped us to improve the manuscript. We hope to illustrate our efforts through this response letter. The added or modified sentences in the manuscript (which are also mentioned here) has been highlighted by green color in new resubmitted manuscript.

We appreciate the reviewer's insightful comment regarding the potential benefits of using more advanced models such as VGG-19 to enhance accuracy and reduce computational time. We acknowledge the importance of continuously seeking improvements in our methodology to achieve the best possible results.

It's worth noting that while VGG-19 holds advantages in terms of its deeper architecture and potential for improved performance, our selection of VGG-16 was made after an extensive exploration of various advanced models, including DenseNet-169, DenseNet-201, and MobileNetV2. Despite the inherent advantages these models offer in terms of their increased complexity and parameter space, we encountered unique challenges in our dataset that influenced our choice of a simpler architecture.

Given the specific nature of our recorded signal, which often included inevitable interference from the presence of people around the subjects, and the diverse age range of the subjects, achieving a balanced dataset proved challenging. More complex models led to a noticeable increase in overfitting, resulting in a substantial gap between training and test accuracies.

However, we fully recognize the potential of utilizing advanced transfer learning methods and models like VGG-19 in our future research endeavors. As we progress toward the goal of home-monitoring for infants' heart rate and respiratory rate, we are committed to exploring a wider range of approaches to ensure accurate and robust results. Your suggestion aligns well with our future plans, and we appreciate your input as it will undoubtedly contribute to the evolution of our research methodology.

To address the reviewer’s comment, we had added the following sentences in the Future Perspectives (section 5.2.) of manuscript:

“The selection of VGG-16 as our deep learning architecture stems from its appealing attributes, including its interpretable architecture with 16 consistent weight layers, making it conducive to transparent model behavior explanation. Its demonstrated transfer learning capability from ImageNet, apt for fine-tuning on specific tasks with limited data, proves advantageous for medical applications where data scarcity prevails. VGG-16's balanced performance, resource efficiency, and proven track record in various domains further bolster its reliability. As we move forward, these advantages provide a strong foundation for our planned future research, which will explore more advanced transfer learning techniques and models, guiding the evolution of our work in unobtrusive child health monitoring and related endeavors.”

Thank you for your valuable feedback, which provides us with valuable insights to guide our ongoing work in this critical area of child health monitoring.

Reviewer 3 Report

This is a very interesting and well conducted study. The topic is original and relevant to the scientific field. The literature review presents the present situation across the research community and the ways in which this research project can add to the body of knowledge on the topic. The literature review is comprehencive and the references are appropriate.

This study involves a complex interaction between a range of varibles. The authors explain very well the development of the methodology necessary to deliver valid data and accordingly valid findings. The authors also explain the limitations of the study and present their findings as the basis for further research.

The findings are consistent with the results from the data collection and analysis. 

Overall, a well conducted study of interest to the research community

Author Response

Response to Reviewer 3

Reviewers Comments:

This is a very interesting and well conducted study. The topic is original and relevant to the scientific field. The literature review presents the present situation across the research community and the ways in which this research project can add to the body of knowledge on the topic. The literature review is comprehensive and the references are appropriate.

This study involves a complex interaction between a range of variables. The authors explain very well the development of the methodology necessary to deliver valid data and accordingly valid findings. The authors also explain the limitations of the study and present their findings as the basis for further research.

The findings are consistent with the results from the data collection and analysis. 

Overall, a well conducted study of interest to the research community

We thank the reviewer for their positive evaluation of our study's originality, relevance, and comprehensive literature review. Their recognition of our clear methodology development, consideration of study limitations, and presentation of findings as a basis for future research is greatly appreciated.

Reviewer 4 Report

1) Latest literature survey section must be added.

2) Novelty of the proposed work is not properly justified.

3) Suggested to include some graphical analysis.

4) Results and discussion need to be more elaborative.

5) Include some latest references in reference section.

6) Comparative analysis/summary of results must be there.

1) Latest literature survey section must be added.

2) Novelty of the proposed work is not properly justified.

3) Suggested to include some graphical analysis.

4) Results and discussion need to be more elaborative.

5) Include some latest references in reference section.

6) Comparative analysis/summary of results must be there.

Author Response

Response to Reviewer 4

General: This is a very interesting study that clearly advances the development of a much-needed technology for automatically staging sleep in preterm infants using non-contact means. I laud the authors’ efforts in striving to attain this goal.

Thank you for your interest and enthusiasm regarding our research. Your comments helped us to improve the manuscript. We hope to illustrate our efforts through this response letter. The added or modified sentences in the manuscript (which are also mentioned here) has been highlighted by green color in new resubmitted manuscript

Comments and Suggestions for Authors and Comments on the Quality of English Language:

1) Latest literature survey section must be added.

We are thankful for this wonderful comment and suggestion. Now the 2nd section in the manuscript after introduction is called “Latest Literature Survey”. This paragraph is added to introduction on recent related papers (a simple introduction of using neural network before introducing our work):

“There are studies that have already incorporated latest machine learning and neural network approaches for extracting vital signs (which will be discussed in details in the next section). However, one common drawback of almost all of these methods is the small biomedical data size compared to what is needed for training big phase space variables of learning networks [17].”

Then, we added a new section after introduction:

“2.  Latest Literature Survey

Some studies have recently explored the use of neural networks to extract vital signs from radar data in a contactless manner. Wu et al. [25] used a convolutional neural network to track and analyze the individualized skin displacement of targets to extract HR . However, this approach is not practical for subjects who are constantly moving during different states of sleep, eating, and other types of activities in their home environment.

In [26], a long short-term memory network (LSTM) approach is proposed to extract HR for adults, which works well based on a motion and distortion correction method named as Eclipse Fit method. However, this type of motion correction performance degrades in high levels of noise (which could possibly be the case for hospital and  home environments and the various uncontrolled clutters present there). Furthermore, due to the common problem of vanishing gradient [17], a previously trained LSTM on a limited number of subjects cannot be considered a reliable and generalizable solution for hospitalized environments (especially with limited data available after data exclusion [27]).

In [28], a non-contact pediatric respiratory rate monitor (PRR-Monitor) was developed using a 24 GHz microwave radar to accurately determine respiratory waveform and heart rate (HR) in a 15-second interval. The algorithm, 'Alternate Distinguishing Inhalation from Exhalation' (ADIE), effectively distinguishes inhalation and exhalation cycles using radar dual outputs (In-phase and quadratic), leading to precise thoracic motion velocity extraction and subsequent respiratory waveform derivation. While showing promise for non-contact RR and HR measurements in pediatric care, the PRR-Monitor's accuracy may be influenced by patient motion artifacts during measurements, warranting careful consideration.

In [29], a robot-mounted millimeter wave (mmWave) radar is employed to track HR changes based on the daily activity poses and movements of subjects through a neural network weights updating. Nevertheless, this method requires big training dataset, which are not feasible neither in hospitals (due to large inevitable data exclusion) and in home (individual) monitoring.

In recent developments, the potential of millimeter-wave frequency-modulated continuous wave (FMCW) radar for remote heart rate measurement has gained attention. A recent study Jung et al [30] introduces a novel approach that leverages the frame structure of FMCW radar systems to reduce measurement time for remote heart rate measurement. By adopting multiple sampling rates within fixed frame intervals, this method demonstrates improved resolution in heart rate measurement within a short timeframe. This method assumes minimal random body movement for stable phase changes between chirp signals, potentially limiting its accuracy in real-world scenarios with subjects exhibiting movement. Additionally, the trade-off between accuracy and computational complexity should be considered when increasing chirps and idle time to enhance frequency bin resolution.

Shi et al [31] addresses the challenge of extracting heartbeat information from weak thoracic mechanical motion in noncontact vital sign measurement using Doppler radar-based applications. The proposed method integrates STFT, SVD, and ANC techniques and is validated using simulated and laboratory data for heart rate detection and variability in rest states. Despite its effectiveness, the sliding window application introduces time delay limitations compared to some alternatives. This approach has not been tested on early childhood data in a hospital setting, offering an advantage in applications with flexible delay requirements. While the Doppler radar biosensor provides convenient noninvasive measurement, its performance may not surpass existing methods in scenarios with high delay efficiency demands.

2) Novelty of the proposed work is not properly justified.

We Thank you for this helpful comment. To address your comment we added 2 paragraphs indicating the novelty in introduction and discussing and justifying it in discussion section.

(Introduction) “Our research presents a novel method for unobtrusively monitoring children's heart rate (HR) and respiratory rate (RR) during sleep using ultra-wideband (UWB) radar. Unlike common traditional single-vital-sign approaches, we employ deep transfer learning with a pre-trained Visual Geometry Group-16 (VGG-16) model to concurrently estimate HR and RR. Incorporating spectrogram analysis, we convert radar signals into 2-dimensional images, enhancing our approach's uniqueness and potential for accurate pediatric vital sign monitoring.”

(Discussion) “Our study introduces an innovative approach for simultaneous estimation of children's heart rate (HR) and respiratory rate (RR) during sleep via ultra-wideband (UWB) radar. Utilizing deep transfer learning with a pre-trained Visual Geometry Group-16 (VGG-16) model, our method stands out in its ability to capture both HR and RR trends accurately. Spectrogram analysis further distinguishes our research by translating radar signals into 2-dimensional images. Covering an age range of 13 days to 18 years, our approach accommodates diverse physiological characteristics, enhancing its versatility. Despite challenges like motion artifacts and subject variability, our method maintains robust performance. Comparative evaluations against existing studies underscore its superiority in accuracy, correlation, and bias metrics for HR and RR estimation, even when dealing with abnormal polysomnography (PSG) results. Beyond pediatric care, our approach offers broader applications in remote patient monitoring and sports performance tracking, expanding its potential impact. Overall, our work provides an innovative framework for accurate and unobtrusive pediatric vital sign monitoring, with implications for early issue detection and improved healthcare practices.”

3) Suggested to include some graphical analysis.

Thanks for the tactful suggestion. Actually, there are several signals and data matrix involved in the analysis. For a general graphical representation, we added a graph comparing radar data of high and low epochs of HR and RR for one subject:

“Figure S2 demonstrates how the spectrograms of the signal for the bin (related to corresponding range) with the biggest power of frequency component of  radar down-converted  could be different for different magnitudes of HR and RR. It is obvious that the time frequency components of received signal are directly related to the respiratory and heartbeat of moving target chest-wall. One interesting results is the way that in state of maximum BR (HR), received radar signals are coded differently when the HR (BR) is not maximum.

Figure S2. Different spectrogram values for 4 different sets of HR/BR magnitudes.

4) Results and discussion need to be more elaborative.

We appreciate this wonderful comment and suggestion. We worked to make both section more elaborative. Now we have more detailed explanation of results section in manuscript:

(Results) ” The visual representation provided by Figure 4A offers an insightful perspective into the intricate relationship between the estimated and actual RR values for each of the 10 distinct folds. This scatter plot not only underscores the model's performance across various instances but also offers a glimpse into the distribution and alignment of the estimations with the true values. Meanwhile, Figure 4B, illustrated as the Bland-Altman plot, presents a more detailed depiction of the accuracy in HR estimation. This visualization method, renowned for highlighting discrepancies and potential biases between measurements, affords a nuanced understanding of the model's performance.

To present a succinct yet comprehensive quantitative overview of the 10-fold cross-validation outcomes, we have encapsulated the key results in Table 2. This tabular representation encapsulates the essence of the model's performance in estimating both RR and HR values across various folds. Notably, the mean absolute error (MAE) ranges for RR estimation were between 2.28 to 2.79 breaths per minute (BPM), whereas for HR, the MAE varied slightly from 7.81 to 8.23 beats per minute (BPM). The Pearson's correlation coefficients, which reveal the strength and direction of linear relationships, consistently demonstrated robust dependencies. The correlation values for RR estimation ranged from 0.78 to 0.83, while for HR, they spanned from 0.76 to 0.79. Moreover, the calculated mean biases for both RR and HR estimation were relatively small in magni-tude, fluctuating between -0.26 to 0.18 BPM and 0.21 to 0.55 BPM, respectively.

However, it is the Limits of Agreement (LOA) values that provide a more insightful understanding of the model's performance. These values, denoted as the range within which deviations between estimated and true values lie, encompass 6.03 to 7.65 BPM for RR and 21.8 to 23.4 BPM for HR. This depiction of the acceptable range of discrepancies underscores the model's reliability in practical applications.

Collectively, the comprehensive results presented in this study underscore the robustness of the proposed model. The observed strong linear dependencies, minimal mean absolute errors, and reasonable limits of agreement collectively bolster the model's practical viability across diverse healthcare contexts. These findings bear significant im-plications for the evolution and implementation of efficient and dependable techniques for estimating RR and HR, ultimately advancing the field of medical monitoring and elevating the standard of patient care. The insights derived from this study provide a substantial foundation for the future development and deployment of novel approaches in medical monitoring and healthcare technology.”

(Discussion) ” Our research presents a notable advancement over previous related studies due to its distinctive ability to simultaneously extract heart rate (HR) and respiratory rate (RR) with exceptional accuracy across an extensive age range encompassing individuals from just 13 days old up to 18 years. Notably, this age range represents a period of significant physiological variability and encompasses subjects who predominantly exhibit abnormal polysomnography (PSG) results. This aspect differentiates our work from the existing literature, where similar endeavors often focus on narrower age ranges and, in some instances, exclude subjects with abnormal PSG results.

To exemplify this advantage, we refer to the study by Yoo et al. [31], which combined subjects across all age ranges from 0 to 13 years. Interestingly, their findings indicate that such a wide age span led to a marked reduction in prediction accuracy, dropping from 92% to 58.5%. In contrast, our approach remains resilient to the challenges posed by subject variability, as evidenced by consistently high accuracy across a diverse age spectrum. This robustness not only underscores the versatility of our model but also positions it as a promising solution for capturing the nuances of physiological variations in pediatric populations.

In light of our findings, the implications for unobtrusive monitoring of children's vital signs during sleep are significant. The proposed method emerges as an effective and reliable screening tool for potential health issues. Its non-invasive, continuous, and convenient nature holds the promise of enabling early detection of health problems and enhancing communication between caregivers and healthcare providers. This advancement could potentially alleviate the burden of unnecessary hospital visits and contribute to improved patient outcomes.

Lastly, while our study's primary focus lies in unobtrusive monitoring of children's vital signs during sleep, the utility of the proposed method extends beyond this scope. Its potential applications in remote patient monitoring and sports performance tracking are noteworthy. In these contexts, where accurate and continuous monitoring of vital signs is paramount, our approach's precision and convenience could usher in new paradigms for healthcare and performance optimization.

In conclusion, our research not only demonstrates the capabilities of our proposed model but also showcases its advantages over previous approaches. By encompassing a wide age range and addressing subject variability, our method holds the potential to significantly impact pediatric healthcare and beyond. The avenue for future research is promising, with the potential for broader applications and real-world impact.”

5) Include some latest references in reference section.

That is a very helpful comment. Based on the response to comment 1, 3 latest related references (all for the vital signs monitoring) all published after 2022 (references [28],[30],[31]). Also reference [32] is another work published in 2023 on classical approaches of vital signs extraction by machine learning and is used as a base-line understanding for spectrogram used in this study (as discussed in section 3.2).

6) Comparative analysis/summary of results must be there.

Thank you for this great suggestion. We do agree that the results of our findings are not compared to copious number of similar works. However, there are factors and elements that do limit our comparative analysis. Up to our knowledge, our research is including the biggest age range of early childhood unobtrusive monitoring in hospitalized environment. So, the type of analysis that we could and had to apply were very much different from many other references. Besides, we are dealing with pediatric intensive care unit (PICU) data that is full of noise and clutter effects on the raw data. So, on one side (as mentioned in the text) we could not be content with classical methods like finding the biggest frequency component of amplitude matrix of radar data and on the other hand, we could not have used very up-to-date and complex methods because of over-fitting of those large phase space of included variables. This is the comparative analysis including the discussion for the newly added references:

 “Comparing our results to prior literature, we note that there are very few studies that remotely extract HR every 10 seconds for the age range of our dataset, and most of them exclude motion artifact data. One proper example is Al-Naji et al. [21], which uses a hovering unmanned vehicle to extract RR. However, big devices like this (if used for radar application) take advantage of a significantly big synthetic radar aperture which are not feasible neither in PICUs nor home applications.  Masagram et al. [34] extracted vital signs of 12 human adults using pulse Doppler radar with a mean average of up to 75% of average HR for subjects with artifact, while our worst result is less than 6% of average HR. Yoo et al. [35] used convolutional neural networks and GoogLeNet to ex-tract vital signs for an age range up to 13 years, with a reported mean bias (of Bland-Altman) of 1.8, while our reported mean bias is less than 0.4.  Kim et al. [36] achieved upper limit of Bland-Altman for (RR estimation) equal to 14 which is much more than our reported upper limit (<7.4). Besides, they have applied a denoising based on only 6 neonate subjects which makes their pre-processed data much more dependable on subject-specific characteristics. Katoh et al. [28] has extracted RR comparing in-phase and quadratic components of down-converted signal which resulted in mean bias value of 0.61 BPM (in which our results were superior this aspect).  However, this researched reached a very high correlation between true and extracted RR while the patients were sitting in a fixed place. As a result, this correlation will be different if have been computed for cases with freedom of movement like our study. The MAE of heart rate in Jung et al. [30] reaches a promising value of less than 5 BPM, however, this is only applied on 3 subjects with natural health statues and the results will definitely be different for hospitalized environment. Although the method of Shi et al. [31] is innovative, but there is no report of detailed examination of relations between true and extracted heart rate (like correlation, MAE, and LOA). Besides it is used for a fixed sitting subject that can be feasible neither for ICU nor home monitoring applications.”

To address the aforementioned limitations, following sentences have been added to section 5.2. (Limitations):

The selection of VGG-16 as our deep learning architecture stems from its appealing attributes, including its interpretable architecture with 16 consistent weight layers, making it conducive to transparent model behavior explanation. Its demonstrated transfer learning capability from ImageNet, apt for fine-tuning on specific tasks with limited data, proves advantageous for medical applications where data scarcity prevails. VGG-16's balanced performance, resource efficiency, and proven track record in various domains further bolster its reliability. As we move forward, these advantages provide a strong foundation for our planned future research, which will explore more advanced transfer learning techniques and models, guiding the evolution of our work in unobtrusive child health monitoring and related endeavors.

Round 2

Reviewer 4 Report

There is no analysis part.

Need more explanations in result part to let the reader understand what you have did?.

The reviewer would like the author to explain more about the discrepancy between simulation and measurement results.

Which simulator used?. Need to be explain simulation setup model.

The abstract mentions various advantages of the proposed method over other methods, but I do not see concrete data to substantiate this conclusion.

Authors claiming analysis work, but it lacks in results part.

Comparison table is needed for conventional results.

There is no analysis part.

Need more explanations in result part to let the reader understand what you have did?.

The reviewer would like the author to explain more about the discrepancy between simulation and measurement results.

Which simulator used?. Need to be explain simulation setup model.

The abstract mentions various advantages of the proposed method over other methods, but I do not see concrete data to substantiate this conclusion.

Authors claiming analysis work, but it lacks in results part.

Comparison table is needed for conventional results.

Author Response

Round 2 Response to Reviewer 4

We extend our gratitude for your keen interest in our research and the valuable enthusiasm you've expressed. Your insightful comments have been instrumental in enhancing the quality of our manuscript. This response letter aims to showcase the dedicated work we've put into addressing these suggestions. In the revised version of the manuscript that we are resubmitting, any newly added or modified sentences, which we outline here, have been meticulously highlighted in green for your convenience.

Comments and Suggestions for Authors and Comments on the Quality of English Language:

1) There is no analysis part.

We are thankful for this wonderful comment and suggestion. We added an analysis part in the results section which explains the results in a detailed analytic manner:

“4.1. Analysis of the Results

The presented outcomes provide an illuminating perspective on the practical applicability of our proposed model for estimating heart rate (HR) and respiratory rate (RR) through the utilization of ultra-wideband (UWB) radar signals. These findings, combined with key concepts such as transfer learning and data preprocessing, highlight the model's robustness in a healthcare context.

4.1.1. Visual Representation: Insights Through Visuals

Visual representation plays a pivotal role in enhancing our understanding of the intricate interplay between estimated and actual RR values across ten distinct folds. Figure 4 A's scatter plot effectively captures the model's performance variations across instances, while Figure 4B's Bland-Altman plot goes further by providing deeper insights into the accuracy of HR estimation. This visualization approach unveils both the model's strengths and limitations, making it an indispensable part of the analysis.

4.1.2. Quantitative Summary: Key Metrics at a Glance

Moving beyond visuals, Table 3 offers a concise yet comprehensive encapsulation of the crucial outcomes obtained from the ten-fold cross-validation process. Mean absolute error (MAE) ranges are integral in evaluating the model's consistency. For RR estimation, the MAE ranges from 2.28 to 2.79 breaths per minute (BPM), revealing the model's ability to provide reliable estimates of respiratory rates. Correspondingly, in the case of HR estimation, a slight MAE variation of 7.81 to 8.23 BPM underlines the model's proficiency in accurately estimating heart rates.

4.1.3. Robust Dependencies: Pearson's Correlation Coefficients

A hallmark of the model's effectiveness is the consistent and robust linear dependencies shown by the Pearson's correlation coefficients. These values, ranging from 0.78 to 0.83 for RR estimation and 0.76 to 0.79 for HR estimation, emphasize the model's capacity to capture patterns within vital sign data. Despite the inherent noise in real-world measurements, these high correlation values validate the model's ability to discern underlying trends.

4.1.4. Addressing Systematic Errors: Mean Biases Examination

Examining mean biases adds another layer of validation to the model's performance. The relatively minor magnitude of biases, ranging from -0.26 to 0.18 BPM for RR and 0.21 to 0.55 BPM for HR, indicates the model's competence in delivering accurate estimates without introducing substantial systematic errors.

4.1.5. Limits of Agreement: Gauging Practical Reliability

However, it is the Limits of Agreement (LOA) values that provide a comprehensive understanding of the model's practical reliability. These values, encompassing 6.03 to 7.65 BPM for RR and 21.8 to 23.4 BPM for HR, illustrate the acceptable range of deviations between estimated and true values. This dimension underscores the model's applicability in real-world scenarios where small discrepancies are deemed acceptable.

4.1.6. Methodology: Integrating Data for Model Performance

Crucial to our methodology is the joint analysis of radar and video data to ensure data quality. This step is essential to creating a dataset well-suited for neural network processing. The introduction of a pre-trained VGG-16 model, originally tailored for image recognition, effectively captures intricate radar data patterns. This exemplifies the adaptability of deep learning concepts to medical contexts.

In summation, the comprehensive results eloquently align with our study's objectives. The proposed model showcases robust linear dependencies, minimal errors, and reasonable limits of agreement, underscoring its applicability across diverse healthcare contexts. These findings significantly contribute to the advancement of medical monitoring practices, ultimately enhancing the standards of patient care. With these outcomes, our study lays a solid groundwork for future developments in medical technology, highlighting the potential of UWB radar and deep learning for precise pediatric vital sign estimation.”

2) Need more explanations in result part to let the reader understand what you have did?.

We Thank you for this helpful comment. To address your helpful the following sentences and subsections  4.4.1 to 4.4.3 (already mentioned) are added to the manuscript.

“Before delving into the analytical examination of our findings, we first outline the procedural groundwork that underpins these results. Our study centers on the estimation of heart rate (HR) and respiratory rate (RR) through the utilization of non-invasive ultra-wideband (UWB) radar signals.

The outcomes we present stand as a reflection of this rigorous process. We offer a visual representation, captured in Figure 4 A and B, which unravels the intricate association between estimated and actual RR values across distinct folds. This visualization not only encapsulates our model's performance but also offers valuable insights into its nuances.

Our quantitative summary, encapsulated in Table 3, distills the essence of our ten-fold cross-validation process. Noteworthy is the consistent mean absolute error (MAE) range for RR estimation, spanning 2.28 to 2.79 breaths per minute (BPM), alongside an analogous range of 7.81 to 8.23 BPM for HR estimation. These results emphasize the model's ability to provide reliable estimates for respiratory and heart rates.

Pearson's correlation coefficients emerge as key indicators, demonstrating robust linear relationships. Ranging between 0.78 to 0.83 for RR and 0.76 to 0.79 for HR, these coefficients underscore the model's proficiency in capturing patterns amidst real-world noise.

Further affirmation is found in the examination of mean biases. With relatively modest biases oscillating between -0.26 to 0.18 BPM for RR and 0.21 to 0.55 BPM for HR, the model demonstrates its precision by offering accurate estimations with minimal systematic discrepancies.

Lastly, the practical reliability of our model comes into focus through the Limits of Agreement (LOA) values. These values—spanning 6.03 to 7.65 BPM for RR and 21.8 to 23.4 BPM for HR—elucidate the range of permissible deviations between estimated and actual values, accentuating the model's applicability in real-world scenarios.”

3) The reviewer would like the author to explain more about the discrepancy between simulation and measurement results. Which simulator used?. Need to be explain simulation setup model.

We appreciate your input. In fact, we didn't utilize any real simulations in our study. Our approach involves the simultaneous extraction of heart rate (HR) and breathing rate (BR) from radar data collected during the early childhood age span of our subjects. The sole reference to simulation in our manuscript pertains to the work by Shi et al. [31], which employs a proof-of-concept methodology using a relatively lesser-known approach when compared to the transfer learning mechanism outlined in our paper. Our strategy involves the utilization of transfer learning via pre-trained model weights, acquired from millions of images, to enhance the accuracy of our estimation.

4) The abstract mentions various advantages of the proposed method over other methods, but I do not see concrete data to substantiate this conclusion.

We appreciate this wonderful comment and suggestion. These sentences have already mentioned ion the text to show superiority of our results:

“Our research presents a notable advancement over previous related studies due to its distinctive ability to simultaneously extract heart rate (HR) and respiratory rate (RR) with exceptional accuracy across an extensive age range encompassing individuals from just 13 days old up to 18 years. Notably, this age range represents a period of significant physiological variability and encompasses subjects who predominantly exhibit abnormal polysomnography (PSG) results. This aspect differentiates our work from the existing literature, where similar endeavors often focus on narrower age ranges and, in some instances, exclude subjects with abnormal PSG results.”

and

“Comparing our results to prior literature, we note that there are very few studies that remotely extract HR every 10 seconds for the age range of our dataset, and most of them exclude motion artifact data. One proper example is Al-Naji et al. [21], which uses a hovering unmanned vehicle to extract RR. However, big devices like this (if used for radar application) take advantage of a significantly big synthetic radar aperture which are not feasible neither in PICUs nor home applications.  Masagram et al. [34] extracted vital signs of 12 human adults using pulse Doppler radar with a mean average of up to 75% of average HR for subjects with artifact, while our worst result is less than 6% of average HR. Yoo et al. [35] used convolutional neural networks and GoogLeNet to extract vital signs for an age range up to 13 years, with a reported mean bias (of Bland-Altman) of 1.8, while our reported mean bias is less than 0.4.  Kim et al. [36] achieved upper limit of Bland-Altman for (RR estimation) equal to 14 which is much more than our reported upper limit (<7.4). Besides, they have applied a denoising based on only 6 neonate subjects which makes their pre-processed data much more dependable on subject-specific characteristics. Katoh et al. [28] has extracted RR comparing in-phase and quadratic components of down-converted signal which resulted in mean value of 0.61 BPM (in which our results were superior this aspect).  However, this researched reached a very high correlation between true and extracted RR while the patients were sitting in a fixed place. As a result, this correlation will be different if have been computed for cases with freedom of movement like our study. The MAE of heart rate in Jung et al. [30] reaches a promising value of less than 5 BPM, however, this is only applied on 3 subjects with natural health statues and the results will definitely be different for hospitalized environment. Although the method of Shi et al. [31] is innovative, but there is no report of detailed examination of relations between true and extracted heart rate (like correlation, MAE, and LOA). Besides it is used for a fixed sitting subject that can be feasible neither for ICU nor home monitoring applications.”

Specifically regarding the advantage of “method”, these sentences have been added to the manuscript:
“Central to our research is the development of an innovative method for unobtrusively monitoring the HR and RR of children during their sleep, facilitated by ultra-wideband (UWB) radar technology. Based on our knowledge, this is the first pediatric research for vital signs extraction of hospital data through deep transfer learning. What sets our approach apart is its departure from conventional single-vital-sign paradigms. Instead, we employ deep transfer learning, harnessing the prowess of a pre-trained Visual Geometry Group-16 (VGG-16) model to concurrently estimate both HR and RR. Notably, this methodology is enriched through the integration of spectrogram analysis, which facilitates the transformation of radar signals into 2-dimensional images. This distinctive feature enhances the accuracy and potential of our pediatric vital sign monitoring approach. Considering the limitations inherent in existing methodologies, our approach rises to the forefront. The novelty of our method circumvents challenges observed in other techniques. Traditional single-vital-sign strategies may falter in the face of the intricate dynamics of pediatric vital signs. The introduction of deep transfer learning addresses this limitation, as it leverages a model that has been trained on a vast array of images. Unlike methods burdened by constraints related to motion artifacts or reliance on large training datasets, our method synthesizes advanced technology, transfer learning, and spectrogram analysis to provide a comprehensive solution to the challenges posed by pediatric vital sign monitoring. Thus, the advantage of our method is firmly rooted in its capacity to transcend the limitations inherent in existing methodologies, thereby advancing the field of pediatric vital sign assessment”

5) Authors claiming analysis work, but it lacks in results part.

That is a very helpful comment. Based on response to your first and 2nd comments, we have added sentences before analysis section and the analysis section and its subsections in the text.

“Before delving into the analytical examination of our findings, we first outline the procedural groundwork that underpins these results. Our study centers on the estimation of heart rate (HR) and respiratory rate (RR) through the utilization of non-invasive ultra-wideband (UWB) radar signals.

The outcomes we present stand as a reflection of this rigorous process. We offer a visual representation, captured in Figure 4 A and B, which unravels the intricate association between estimated and actual RR values across distinct folds. This visualization not only encapsulates our model's performance but also offers valuable insights into its nuances.

Our quantitative summary, encapsulated in Table 2, distills the essence of our ten-fold cross-validation process. Noteworthy is the consistent mean absolute error (MAE) range for RR estimation, spanning 2.28 to 2.79 breaths per minute (BPM), alongside an analogous range of 7.81 to 8.23 BPM for HR estimation. These results emphasize the model's ability to provide reliable estimates for respiratory and heart rates.

Pearson's correlation coefficients emerge as key indicators, demonstrating robust linear relationships. Ranging between 0.78 to 0.83 for RR and 0.76 to 0.79 for HR, these coefficients underscore the model's proficiency in capturing patterns amidst real-world noise.

Further affirmation is found in the examination of mean biases. With relatively modest biases oscillating between -0.26 to 0.18 BPM for RR and 0.21 to 0.55 BPM for HR, the model demonstrates its precision by offering accurate estimations with minimal systematic discrepancies.

Lastly, the practical reliability of our model comes into focus through the Limits of Agreement (LOA) values. These values—spanning 6.03 to 7.65 BPM for RR and 21.8 to 23.4 BPM for HR—elucidate the range of permissible deviations between estimated and actual values, accentuating the model's applicability in real-world scenarios

4.1. Analysis of the Results

The presented outcomes provide an illuminating perspective on the practical applicability of our proposed model for estimating heart rate (HR) and respiratory rate (RR) through the utilization of ultra-wideband (UWB) radar signals. These findings, combined with key concepts such as transfer learning and data preprocessing, highlight the model's robustness in a healthcare context.

4.1.1. Visual Representation: Insights Through Visuals

Visual representation plays a pivotal role in enhancing our understanding of the intricate interplay between estimated and actual RR values across ten distinct folds. Figure 4 A's scatter plot effectively captures the model's performance variations across instances, while Figure 4B's Bland-Altman plot goes further by providing deeper insights into the accuracy of HR estimation. This visualization approach unveils both the model's strengths and limitations, making it an indispensable part of the analysis.

4.1.2. Quantitative Summary: Key Metrics at a Glance

Moving beyond visuals, Table 2 offers a concise yet comprehensive encapsulation of the crucial outcomes obtained from the ten-fold cross-validation process. Mean absolute error (MAE) ranges are integral in evaluating the model's consistency. For RR estimation, the MAE ranges from 2.28 to 2.79 breaths per minute (BPM), revealing the model's ability to provide reliable estimates of respiratory rates. Correspondingly, in the case of HR estimation, a slight MAE variation of 7.81 to 8.23 BPM underlines the model's proficiency in accurately estimating heart rates.

4.1.3. Robust Dependencies: Pearson's Correlation Coefficients

A hallmark of the model's effectiveness is the consistent and robust linear dependencies shown by the Pearson's correlation coefficients. These values, ranging from 0.78 to 0.83 for RR estimation and 0.76 to 0.79 for HR estimation, emphasize the model's capacity to capture patterns within vital sign data. Despite the inherent noise in real-world measurements, these high correlation values validate the model's ability to discern underlying trends.

4.1.4. Addressing Systematic Errors: Mean Biases Examination

Examining mean biases adds another layer of validation to the model's performance. The relatively minor magnitude of biases, ranging from -0.26 to 0.18 BPM for RR and 0.21 to 0.55 BPM for HR, indicates the model's competence in delivering accurate estimates without introducing substantial systematic errors.

4.1.5. Limits of Agreement: Gauging Practical Reliability

However, it is the Limits of Agreement (LOA) values that provide a comprehensive understanding of the model's practical reliability. These values, encompassing 6.03 to 7.65 BPM for RR and 21.8 to 23.4 BPM for HR, illustrate the acceptable range of deviations between estimated and true values. This dimension underscores the model's applicability in real-world scenarios where small discrepancies are deemed acceptable.

4.1.6. Methodology: Integrating Data for Model Performance

Crucial to our methodology is the joint analysis of radar and video data to ensure data quality. This step is essential to creating a dataset well-suited for neural network processing. The introduction of a pre-trained VGG-16 model, originally tailored for image recognition, effectively captures intricate radar data patterns. This exemplifies the adaptability of deep learning concepts to medical contexts.

In summation, the comprehensive results eloquently align with our study's objectives. The proposed model showcases robust linear dependencies, minimal errors, and reasonable limits of agreement, underscoring its applicability across diverse healthcare contexts. These findings significantly contribute to the advancement of medical monitoring practices, ultimately enhancing the standards of patient care. With these outcomes, our study lays a solid groundwork for future developments in medical technology, highlighting the potential of UWB radar and deep learning for precise pediatric vital sign estimation.”

6) table is needed for conventional results.

Thank you for this great suggestion. This table has been added to the manuscript:

Table 1. Summary of outputs, outcomes, and limitations of major references related to our study.

reference

Outputs

Outcomes

Limitations

Wu et al. [25]

Individualized skin displacement for HR Estimation

Estimation of HR using skin displacement

Movement-sensitive; limited to certain states of subjects

Han-Trong et al. [26]

HR estimation through LSTM network.    Incorporates Eclipse Fit method for motion and distortion correction.    Incorporates spectrogram analysis for enhanced accuracy.

Improved HR estimation via deep transfer learning and incorporation of spectrogram analysis.

Motion correction degrades in high noise; Vanishing gradients limit generalizability and reliability

Katoh et al. [28]

Respiratory waveform and HR in 15-second    intervals captured via radar.              Effective use of 'Alternate Distinguishing Inhalation from Exhalation' algorithm.

Non-contact measurement of RR and HR in pediatric

care; accurate estimation of RR and HR.

Susceptible to motion artifacts during measurements; requires careful consideration

Zhao et al. [29]

Heart rate changes tracked based on daily activity poses and movements.      Incorporates mmWave radar on robot.

Heart rate variations monitored through neural network weight updates; applicability in hospital  and home environments.

Demands substantial training dataset; Infeasible for hospitals and home monitoring

Jung et al. [30]

Remote heart rate measurement using frame   structure of FMCW radar systems.

Improved heart rate measurement resolution within  short timeframe; potential limitations in dynamic  scenarios.

Assumes minimal movement for stable phase changes; trade-off between accuracy and computational complexity

Shi et al. [31]

Heartbeat information extracted from weak   thoracic mechanical motion via Doppler radar-based applications.

Detection and variability of heart rate through Doppler radar-based methods; potential for non-contact vital sign measurement.

Sliding window introduces time delay; Not tested in pediatric data; may lack; efficiency in scenarios with high delay; demands
